# Integrated and Metal Free Synthesis of Dimethyl Carbonate and Glycidol from Glycerol Derived 1,3-Dichloro-2-propanol via CO$_2$ Capture

**Santosh Khokarale [1,\*], Ganesh Shelke [1] and Jyri-Pekka Mikkola [1,2,\*]**

[1]  Technical Chemistry, Chemical-Biological Centre, Department of Chemistry, Umeå University, SE-90187 Umeå, Sweden; ganesh.shelke@umu.se
[2]  Industrial Chemistry & Reaction Engineering, Johan Gadolin Process Chemistry Centre, Department of Chemical Engineering, Åbo Akademi University, FI-20500 Åbo-Turku, Finland
\*  Correspondence: santosh.khokarale@umu.se (S.K.); jyri-pekka.mikkola@umu.se (J.-P.M.)

**Abstract:** Dimethyl carbonate (DMC) and glycidol are considered industrially important chemical entities and there is a great benefit if these moieties can be synthesized from biomass-derived feedstocks such as glycerol or its derivatives. In this report, both DMC and glycidol were synthesized in an integrated process from glycerol derived 1,3-dichloro-2-propanol and CO$_2$ through a metal-free reaction approach and at mild reaction conditions. Initially, the chlorinated cyclic carbonate, i.e., 3-chloro-1,2-propylenecarbonate was synthesized using the equivalent interaction of organic superbase 1,8-diazabicyclo [5.4.0] undec-7-ene (DBU) and 1,3-dichloro-2-propanol with CO$_2$ at room temperature. Further, DMC and glycidol were synthesized by the base-catalyzed transesterification of 3-chloro-1,2-propylenecarbonate using DBU in methanol. The synthesis of 3-chloro-1,2-propylenecarbonate was performed in different solvents such as dimethyl sulfoxide (DMSO) and 2-methyltetrahydrofuran (2-Me-THF). In this case, 2-Me-THF further facilitated an easy separation of the product where a 97% recovery of the 3-chloro-1,2-propylenecarbonate was obtained compared to 63% with DMSO. The use of DBU as the base in the transformation of 3-chloro-1,2-propylenecarbonate further facilitates the conversion of the 3-chloro-1,2 propandiol that forms in situ during the transesterification process. Hence, in this synthetic approach, DBU not only eased the CO$_2$ capture and served as a base catalyst in the transesterification process, but it also performed as a reservoir for chloride ions, which further facilitates the synthesis of 3-chloro-1,2-propylenecarbonate and glycidol in the overall process. The separation of the reaction components proceeded through the solvent extraction technique where a 93 and 89% recovery of the DMC and glycidol, respectively, were obtained. The DBU superbase was recovered from its chlorinated salt, [DBUH][Cl], via a neutralization technique. The progress of the reactions as well as the purity of the recovered chemical species was confirmed by means of the NMR analysis technique. Hence, a single base, as well as a renewable solvent comprising an integrated process approach was carried out under mild reaction conditions where CO$_2$ sequestration along with industrially important chemicals such as dimethyl carbonate and glycidol were synthesized.

**Keywords:** carbon dioxide; dimethyl carbonate; glycidol; organic superbase; integrated synthesis

## 1. Introduction

Glycerol is a highly precious and industrially important biomass-derived molecule since it has numerous applications in the pharmaceuticals, cosmetics, and food industries [1]. Besides that, it is also serving a vital role in the production of various low molecular weight commodity chemicals, e.g., ethylene glycol, 1,2- and 1,3-propanediol, acrylic acid, glycerol carbonate, glyceraldehyde, dihydroxyacetone, etc. [2–4]. Glycerol is a co-product of the biodiesel synthesis process and since the production of biodiesel increased tremendously, the glycerol is also produced in huge amounts simultaneously [3].

Albeit, excess glycerol is often disposed of as a waste, it is not economically beneficial for the biodiesel industries considering the overall cost of the process as well as the negligible value addition to such a vital and renewable chemical. Hence, it is necessary to establish more efficient and alternative pathways to utilize glycerol, for example in the synthesis of value-added chemicals, fuel additives, etc. Considering the excellent source of $C_3$ carbon backbone, glycerol is used to produce lactic acid, carbonates (linear and cyclic), diols, esters, and epichlorohydrin (ECH), which further reduces the dependency on fossil-derived routes upon their production [5,6].

The chlorination of the liquid glycerol to di-chlorinated analogies such as 2,3-dichloro-1-propanol and 1,3-dichloro-2-propanol is a well industrially applied process [5–7] (Scheme 1). This process is a part of Solvay's Epicerol process, which is applied for the synthesis of industrially important ECH (epoxy resin monomer) where the annual production has reached more than 100 kt [8,9]. This process not only replaced the traditional method for the synthesis of ECH, such as the chlorination of propene at high temperatures, but also increased the renewable nature of ECH since the processes use glycerol as one of the reagents in the synthesis.

**Scheme 1.** Epichlorohydrin synthesis from glycerol (Solvay's Epicerol process).

As shown in Scheme 1, the chlorination of glycerol is carried out with two moles of hydrochloric acid using Lewis acid catalysts such as carboxylic acid (e.g., acetic acid) to form a mixture of 2, 3-dichloro-1-propanol and 1,3-dichloro-2-propanol. Further, out of these chlorinated derivatives of glycerol, 1,3-dichloro-2-propanol is converted to ECH following the alkaline hydrolysis process [7]. The synthesis of these chlorinated analogues of the glycerol and their further application is only limited to the ECH synthesis, whereas these analogues are not explored for other fruitful applications. 2-chloro-1,3-propanol, one of the mono-chlorinated analogues of glycerol, is considered as a waste in the Epicerol process. In this case, 2-chloro-1,3-propanol cannot be converted to the di-chlorinated species because the chlorine at the beta position (β form) inhibits further chlorination. Proto et al. proposed that 2-chloro-1,3-propanol can be converted to glycidol, which is also considered a highly active and vital chemical entity in polymer, rubber, as well as in dye industries [7]. In other words, identical to the synthesis of ECH from glycerol, the processing of 2-chloro-1,3-propanol for glycidol synthesis can emerge as a new alternative for the existing allyl alcohol epoxidation using an $H_2O_2$ precursor and titanium silicate catalyst, TS-1 [10]. Hence, this integrated approach for the synthesis of ECH and glycidol can increase the atom economy as well as the overall sustainability of the Epicerol process. However, besides synthesis of the ECH and glycidol, it is necessary to implement more available applications of the chlorinated analogs of the glycerol to enhance the applicability of a surplus amount of glycerol from the biodiesel industries.

In this work, we report the integrated method for the valorization of 1,3-dichloro-2-propanol to dimethyl carbonate (DMC) and glycidol through an organic superbase involving $CO_2$ capture and a base-catalyzed process. Being less toxic as well as having versatile reactivity, the use of DMC as a reagent as well as a solvent in various organic transformations is considered a green, sustainable, and environmentally friendly approach. Besides having combined the functionality of $CO_2$ and the methyl group, DMC is success-fully utilized for the valorization of bio-based building blocks to value-added chemicals and fuels as well as for the derivatization of cellulose-to-cellulose methyl carbonate [11,12]. Considering the vital role of DMC in synthetic chemistry, several catalytic processes with

and without the use of $CO_2$ have been developed for the DMC synthesis where some of the methods have been commercialized [13,14].

In this reaction approach, the $CO_2$ molecule was initially activated through the equivalent interaction between 1,3-dichloro-2-propanol and organic superbase diazabicyclo [5.4.0] undec-7-ene (DBU), where the resultant 3-chloro-1,2-propylenecarbonate was further transesterified with methanol to form DMC and glycidol (Scheme 2).

**Scheme 2.** (**a**) Synthesis of 3-chloro-1,2-propylenecarbonate from 1,3-dichloro-2-propanol and, (**b**) Transesterification of 3-chloro-1,2-propylenecarbonate to dimethyl carbonate and glycidol.

The DBU superbase ($pk_a = 23$)-mediated activation of $CO_2$ is a well-studied process where it not only emerged as a new and greener pathway for the up-gradation of $CO_2$, but it also introduced new reversible solvent media called switchable ionic liquid (SIL), which was efficiently used for the processing of lignocellulosic biomass such as wood and cellulose esters synthesis [15,16]. In this actual case, DBU initially deprotonates alcohols (R-OH) where the resultant alkoxide anion equivalently reacts with $CO_2$ to form [DBUH][ROCO_2] salt in the form of an SIL [17,18]. Besides the $CO_2$ capture, the synthesis of SIL has also been further explored upon the synthesis of linear as well as cyclic carbonates, methyl formate, as well as acrylic plastic precursors synthesis [19–22]. In this regard, the synthesis of cyclic carbonates using 1,2 chlorohydrins has been also previously reported for the synthesis of various cyclic carbonates [20]. Similar work of the synthesis of cyclic carbonates now mimicked in this report in the case of the synthesis of the 3-chloro-1,2-propylenecarbonate where 1,3-dichloro-2-propanol is assumed as the 1,2 chlorohydrin. After the complete synthesis, the recovery method has also been further set up for the separation of DMC and glycidol following solvent extraction techniques. In addition, as shown in Scheme 2, the overall process of the synthesis of DMC and glycidol was also accompanied by the formation of [DBUH][Cl] salt, which was further separated from the reaction mixture and further used for the recovery of DBU. The progress of the reaction as well as the purity of the recovered chemical species was confirmed by means of NMR analysis techniques.

## 2. Materials and Methods
### 2.1. Chemicals and Methods
#### 2.1.1. Chemicals

1,3-dichloro-2-propanol (98%), 1,8-Diazabicyclo[5.4.0]undec-7-ene (DBU, 98%), dimethyl carbonate (DMC), glycidol (96%), $D_2O$ (99.9 atom % D), and CDCl3 (99.8 atom % D) were purchased from Sigma Aldrich (Saint Louis, MO, USA), whereas methanol ($\geq$99.0%), dimethyl sulfoxide ($\geq$99.0%) and 2-Methyltetrahydrofuran (2-Me-THF, biorenewable, anhydrous, $\geq$99%, Inhibitor-free) were purchased from VWR chemicals and used without further purification. The $CO_2$ gas bottle was supplied by AGA AB (Linde Group) and used without further purification.

### 2.1.2. NMR Analysis

The progress during the synthesis of DMC and glycidol in the process was confirmed by means of NMR analysis using Bruker Avance 400 MHz instrument (Billerica, MA, USA). The CDCl₃ or capillary filled with D₂O was used as an internal standard during the analysis. The obtained data were further processed with TopSpin 4.0.7 software (Billerica, MA, USA). After NMR analysis, the types of chemical species observed during the synthesis are shown in Figure 1.

**Figure 1.** Chemical species formed during the synthesis. (The * and # signs used to highlight carbonyl carbon in 3-chloro-1,2-propylenecarbonate and dimethyl carbonate, respectively in NMR spectra).

### 2.2. Synthesis of 3-Chloro-1,2-propylenecarbonate in DMSO or 2-Me-THF

The synthesis of 3-chloro-1,2-propylenecarbonate was carried out either in DMSO or 2-Me-THF. In this case, initially, 0.63 g (4.9 mmol) of 1,3-dichloro-2-propanol was mixed with 4 mL of DMSO solvent under stirring and the reaction flask was kept in a water bath (21 °C) for 10 min. Then, the CO₂ gas (100 mL/min) was bubbled at room temperature in the reaction mixture for 5 min followed by dropwise addition of 0.75 mL (4.9 mmol) of DBU carried out at water bath temperature. The CO₂ gas was bubbled for a further 10 min in the reaction mixture to ensure complete interaction between added reagents. In the case of 2-Me-THF as a solvent, a similar process was applied during the synthesis of 3-chloro-1,2-propylenecarbonate where similar amounts of all the reagents have been used.

### 2.3. Recovery of 3-Chloro-1,2-propylenecarbonate

Solvent extraction was used to separate the various reaction components formed during the 3-chloro-1,2-propylenecarbonate synthesis. In this case, before applying the recovery method, the complete conversion of the reagents was initially confirmed with NMR analysis. The DMSO solvent comprised reaction mixture was added to 20 mL of ethyl acetate under stirring where white solid belonged to the chloride salt of DBU, i.e., [DBUH][Cl] was precipitated out. The [DBUH][Cl] salt was separated by filtration from the reaction mixture and washed with 15 mL of ethyl acetate. The DBU salt was further vacuum dried at 40 °C and stored in a desiccator before its purity confirmation using NMR analysis. The DMSO containing organic phase was added to 20 mL of water where DMSO was extracted with water while 3-chloro-1,2-propylenecarbonate remained in the ethyl acetate phase. Both phases were separated using a separating funnel. Further, the organic phase was concentrated on a rotary evaporator to obtain 3-chloro-1,2-propylene carbonate after drying over anhydrous sodium sulfate. In the case of 2-Me-THF solvent containing reaction mixture, the solvent was removed from the reaction mixture by a rotary evaporator. Further, ethyl acetate was added to the reaction mixture where precipitated [DBUH][Cl] salt and 3-chloro-1,2-propylenecarbonate with ethyl acetate were separated using filtration, rotatory evaporator, and vacuum drying methods. Recovery of the 3-chloro-1,2-propylenecarbonate with various solvents was calculated using equation 1, where the

theoretical amount of the 3-chloro-1,2-propylenecarbonate was calculated based on the initial moles of 1,3-dichloro-2-propanol used in the synthesis.

$$\% \text{ Recovery of } 3-Cl-1,2-\text{propylenecarbonate} = \frac{\text{Recovered } 3-Cl-1,2-\text{propylenecarbpnate (moles)} \times 100}{\text{Theriotical amount of } 3-Cl-1,2-\text{propylenecarbonate (moles)}} \quad (1)$$

### 2.4. Transesterification of 3-Chloro-1,2-propylenecarbonate to Synthesis DMC and Glycidol and Their Recoveries

The synthesis of DMC and glycidol from 3-chloro-1,2-propylenecarbonate was performed in methanol and studied at various temperatures. Equivalent amounts of 3-chloro-1,2-propylenecarbonate and DBU were mixed with methanol and the reaction mixture was stirred at room temperature (22 °C), 35 or 50 °C for various reaction times. At end of the reaction, methanol with DMC was separated from the reaction mixture by high vacuum distillation at 40 °C. The amount of DMC recovered with methanol was confirmed using gas chromatography where the calibration method and Equation number (2) were used to calculate the actual amount of DMC recovered (supporting information Figure S1). Glycidol and [DBUH][Cl] salt were separated from each other using 2-Me-THF and brine solution (NaCl saturated aqueous solution) as extracting solvents. In this case, 2-Me-THF and brine solution were added to the [DBUH][Cl] salt and glycidol mixture where the organic phase was separated from the aqueous phase using a separating funnel. For the glycidol recovery, the organic phase was dried over anhydrous sodium sulfate and concentrated by rotary evaporation. The recovery of glycidol was calculated using Equation (3). The aqueous phase was concentrated by rotation evaporation where dry methanol was added further in [DBUH][Cl] salt and NaCl mixture to precipitate the NaCl. The NaCl salt was separated from the alcoholic solution by filtration, whereas the [DBUH][Cl] salt was recovered after the alcoholic solution by rotary evaporation. The purity of the recovered [DBUH][Cl] salt and glycidol was confirmed using NMR analysis.

$$\% \text{ Recovery of dimethyl carbonate (DMC)} = \frac{\text{Recovered DMC (moles)} \times 100}{\text{Theortical amount of DMC (moles)}} \quad (2)$$

$$\% \text{ Recovery of Glycidol} = \frac{\text{Recovered glycidol (moles)} \times 100}{\text{Theoretical amount of glycidol (moles)}} \quad (3)$$

### 2.5. Recovery of DBU from [DBU][Cl] Salt Using a Neutralization Method

A total of 0.12 g (3 mmol) of NaOH was added to 10 mL of dry methanol, the reaction mixture was stirred at 50 °C for 1 h, and a transparent solution was obtained. Further, 0.47 g (2.5 mmol) of [DBUH][Cl] salt was added to this alkaline methanol solution where the reaction mixture was stirred at 50 °C for 1 h. As the reaction progressed, a white and crystalline precipitate of NaCl separated and settled at the bottom of the reaction flask. The NaCl salt was separated from the reaction mixture by filtration and DBU was recovered from the alcoholic solution by rotary evaporation. The recovery degree of DBU was calculated by using Equation (4), whereas the purity was confirmed using NMR analysis. To calculate the amount of recovered DBU, the theoretical amount of DBU was calculated based on the initial amount of [DBUH][Cl] salt (moles) that was used in the recovery process.

$$\% \text{ Recovery of DBU} = \frac{\text{Recovered DBU (moles)} \times 100}{\text{Theoretical amount of DBU (moles)}} \quad (4)$$

### 3. Results

DMC and glycidol synthesis proceeded via the integrated two-step process approach. In this case, initially, the synthesis of 3-chloro-1,2-propylenecarbonate was prepared through the equivalent interaction of 1,3-dichloro-2-propanol, DBU, and $CO_2$, at room

temperature. Further, the DMC along with glycidol were synthesized via a base-catalyzed transesterification of 3-chloro-1,2-propylenecarbonate in methanol. The synthesis of 3-chloro-1,2-propylenecarbonate was initially carried out in DMSO as a solvent and a similar synthesis was further studied in other solvents such as 2-Me-THF. After the complete addition of DBU in the reaction mixture containing DMSO and 1,3-dichloro-2-propanol under $CO_2$ bubbling, the composition of the resultant reaction mixture was confirmed by one- as well as two-dimensional NMR analysis. The $^1H$ and $^{13}C$ NMR spectra of the reaction mixture are shown in Figure 2 and supporting information Figure S2, respectively. As shown in Figure 2, after the interaction between 1,3-dichloro-2-propanol, DBU, and $CO_2$, DBU as well as 1,3-dichloro-2-propanol were completely consumed in the reaction mixture as their corresponding signals for the carbon atoms disappeared. In this case, the characteristics signals for the carbon atoms at positions six, seven, and nine, respectively, in the molecular DBU disappeared and new shielded signals for the carbon atoms at position six and seven (C-6′ and C-7′) as well as a de-shielded signal for carbon atom at position nine (C-9′), respectively, were observed.

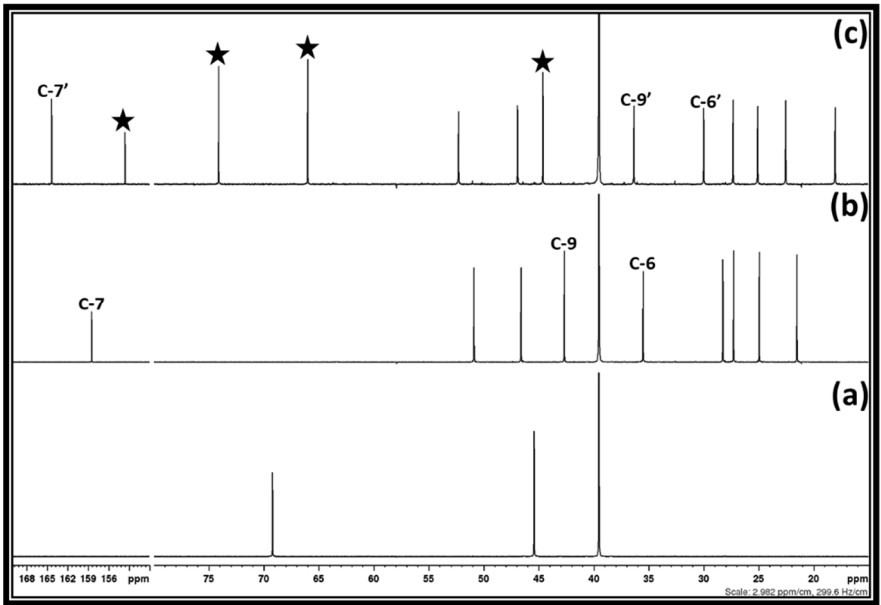

**Figure 2.** $^{13}C$ NMR spectra of the (**a**) 1,3-dichloro-2-propanol, (**b**) DBU, and (**c**) Reaction mixture after equivalent interaction of 1,3-dichloro-2-propanol, DBU, and $CO_2$ in DMSO (NMR analysis with capillary filled with $D_2O$).

This observation represents that the $sp^2$-N atom in the DBU molecule became protonated, which is in agreement with the previous reports [20]. Besides the signals for the protonated DBU, the signals for the unknown chemical species were also observed in the $^{13}C$ NMR analysis (shown by a filled star). $^1H$ NMR spectra also depict that the characteristics signals for the protons in both 1,3-dichloro-2-propanol and DBU molecules, respectively, disappeared, while signals for the unknown chemical species as well as protonated DBU appeared. As described previously, the DBU molecule is popularly known as a superbase to activate the $CO_2$ molecule through the formation of SIL in the presence of proton sources such as water or alcohol. Besides that, it was also previously confirmed that the equivalent interaction between 1, 2-halohydrin and DBU molecule in the presence of $CO_2$ results in the formation of cyclic carbonate [20]. Since 1,3-dichloro-2-propanol molecule has a similar structure to the 1,2-halohydrin, i.e., –OH and halide groups are attached to the adjacent carbon atoms, its interaction with DBU and $CO_2$ molecules could result in the formation of cyclic carbonate such as 3-chloro-1,2-propylenecarbonate. To confirm the formation of 3-chloro-1,2-propylenecarbonate, the reaction mixture obtained was analyzed using two dimensional (2D) HMBC (Heteronuclear Multiple Bond Correlation),

HSQC (Heteronuclear Single Quantum Coherence), and COSY (Correlated Spectroscopy) NMR analysis techniques and corresponding spectra are shown in Figure 3 and supporting information is shown in Figure S3a,b.

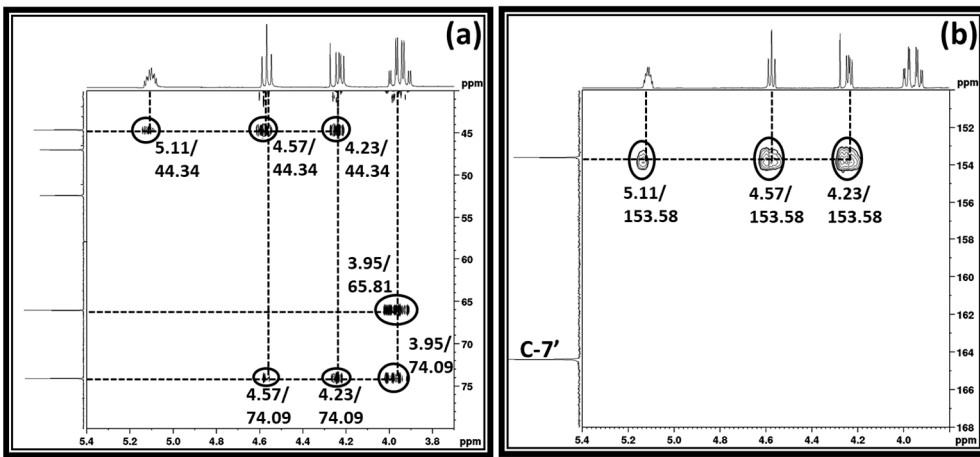

**Figure 3.** (**a**,**b**) $^1$H-$^{13}$C HMBC NMR spectra of the reaction mixture after equivalent interaction of 1,3-dichloro-2-propanol, DBU and $CO_2$ in DMSO.

The HSQC NMR analysis shows that the protons in an unknown chemical species with chemical shifts 3.95, 4.23, 4.57, and 5.11 ppm, respectively, belong to the proton–carbon correlation signals with their corresponding carbon atoms (supporting information S3a). In the case of COSY NMR analysis, the proton with the chemical shift 5.11 ppm proton–proton correlated with all the remaining protons, whereas the protons at 4.23 and 4.57 ppm did not show any correlation with the proton resonating at 3.95 ppm (supporting information S3b). This suggests that the distribution of the protons in this unknown chemical species is identical to the 1,3-dichloro-2-propanol that was used in the synthesis. The HMBC NMR analysis showed that protons with chemical shifts 4.23, 4.57, and 5.11, respectively, are in correlation with the carbon atom resonating at 153.5 ppm. The signal for the carbon atom at 153.5 ppm was a new one and it usually belongs to the carbon atom in a carbonyl group. This observation suggests that the activation of the $CO_2$ molecule took place through the equivalent interaction between the reagents applied in the synthesis. Since the [DBUH]$^+$ cation forms in the reaction composition, the formation of this cation proceeds through the activation of $CO_2$ by the DBU superbase. In this case, as shown in Scheme 3a, DBU removed the proton from the –OH group in 1,3-dichloro-2-propanol and the resultant alkoxide species reacted with $CO_2$ and alkyl carbonate anion, whereupon the [DBUH]$^+$ cation formed in the reaction mixture. However, since the COSY and HMBC NMR analysis suggests that the protons with the chemical shifts 4.23, 4.57, and 5.11, respectively, are adjacent to each other and in long correlation with 153.5 ppm, further consecutive cyclization in alkyl carbonate anion took place, which further allowed the formation of 3-chloro-1,2-propylenecarbonate along with the release of [DBUH][Cl] salt (Scheme 3b). Hence, similar to the previously reported cyclic carbonate synthesis, the equivalent interaction between 1,3-dichloro-2-propanol, DBU, and $CO_2$ results in a 3-chloro-1,2-propylenecarbonate, which formed in the process [20]. This DBU mediated fixation of $CO_2$ in the form of 3-chloro-1,2-propylenecarbonate was carried out at room temperature and atmospheric pressure. Therefore, this method can be considered safer and sustainable compared to epichlorohydrin encompassed with high energy-consuming catalytic approaches. Even though both 1,3-dichloro-2-propanol and epichlorohydrin are derived from the hydro-chlorination of glycerol, the processing with epichlorohydrin in 3-chloro-1,2-propylenecarbonate synthesis is not safe considering its toxic and flammable nature.

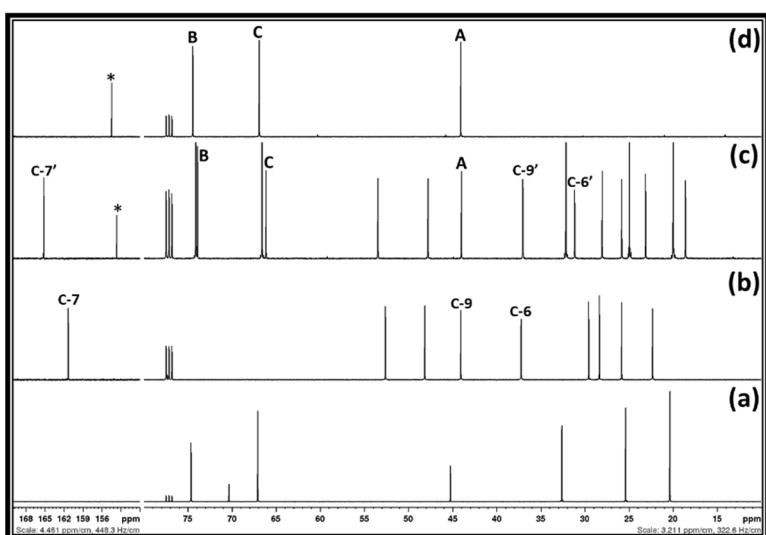

**Scheme 3.** (**a**) Activation of $CO_2$ by DBU through proton abstraction from 1,3-dichloro-2-propanol and (**b**) formation of 3-chloro-1,2-propylenecarbonate and [DBUH][Cl] salt.

After the synthesis of 3-chloro-1,2-propylenecarbonate, the reaction mixture was further explored in terms of its recovery via water and ethyl acetate-involved solvent extraction methods. In this case, after the separation of [DBUH][Cl] salt using ethyl acetate, water was further used to separate DMSO from 3-chloro-1,2-propylenecarbonate, which remained in the organic phase. After the removal of ethyl acetate, a 68% recovery of 3-chloro-1,2-propylenecarbonate was achieved. This represents that even though 3-chloro-1,2-propylenecarbonate is insoluble in water due to DMSO, a part of it remained in the aqueous phase. Further, a similar synthesis of 3-chloro-1,2-propylenecarbonate was carried out in 2-Me-THF solvent. During the bubbling of $CO_2$ and the simultaneous addition of DBU in the reaction mixture, containing 1,3-dichloro-2-propanol in a 2-Me-THF, it was observed that a white crystalline solid precipitate was forming and became separated in the reaction mixture. The $^{13}C$ NMR analysis of the reaction mixture along with the white precipitate was carried out and the obtained spectra are shown in Figure 4.

**Figure 4.** $^{13}C$ NMR spectra of the (**a**) 1,3-dichloro-2-propanol in 2-Me-THF, (**b**) DBU and, (**c**) Reaction mixture after equivalent interaction of 1,3-dichloro-2-propanol, DBU, and $CO_2$ in 2-Me-THF, and (**d**) 3-chloro-1,2-propylenecarbonate (NMR analysis in CDCl$_3$).

Figure 4 depicts that the 3-chloro-1,2-propylenecarbonate as well as [DBUH][Cl] salt were forming after an equivalent interaction between 1,3-dichloro-2-propanol, DBU, and $CO_2$ when 2-Me-THF was used as the solvent during the synthesis. Recently, Jupke et al.

reported that 2-Me-THF has a higher $CO_2$ solubility in the reaction system than water under identical experimental conditions [23]. Jessop and Matsuda et al. showed that 2-Me-THF has lower values of Kamlet-Taft parameters such as polarizability, $\pi^*$ (0.5–1.1), which further allowed for the higher solubility of hydrophobic $CO_2$ in 2-Me-THF than water [24,25]. Matsuda et al. also further reported that $CO_2$ expanded 2-Me-THF, and other bio-based solvents turned out to be an excellent solvent media for biotransformation. The author explained that with an increase in the $CO_2$ pressure, the polarizability ($\pi^*$) value of the 2-Me-THF linearly decreased as a result of the higher solubility of $CO_2$, which further increased the transformation rate in this $CO_2$ expanded solvent system [26].

Therefore, similar to DMSO, 2-Me-THF can be used as a solvent in the synthesis of 3-chloro-1,2-propylenecarbonate. In this case, 2-Me-THF is preferred more considering its renewable nature and this solvent is already referred to as an alternative to THF and other organic solvents [27,28]. After the completion of the reaction, 2-Me-THF was further removed by evaporation and [DBUH][Cl] salt was separated from 3-chloro-1,2-propylenecarbonate using ethyl acetate solvent and a filtration technique. Further, 93% of the 3-chloro-1,2-propylenecarbonate was recovered when ethyl acetate was removed from the organic phase. Hence, the use of 2-Me-THF in the synthesis not only facilitated the separation of components from the reaction mixture but also further allowed a higher level of recovery of 3-chloro-1,2-propylenecarbonate.

To valorize 3-chloro-1,2-propylenecarbonate, it was further explored in the base-catalyzed transesterification in methanol where DBU was used as a base and the synthesis was carried out at different temperatures. The current catalytic approaches for the synthesis of 3-chloro-1,2-propylenecarbonate from ECH are considered as only an ideal example to demonstrate the valorization of $CO_2$ upon the synthesis of cyclic carbonates. Hence, 3-chloro-1,2-propylenecarbonate remains underutilized and needs to be upgraded to value-added chemical entities considering the value of the active form of $CO_2$ in the molecule. The concept of the transesterification of 3-chloro-1,2-propylenecarbonate under alkaline conditions was designed based on the previous reports where the cyclic carbonates such as ethylene carbonates, catechol carbonate, etc. were used to synthesize various aliphatic carbonates [13,29]. As shown in Scheme 4, the reaction of the alkaline transesterification involved the interaction of methanol with 3-chloro-1,2-propylenecarbonate and results in the formation of the DMC and 3-Chloro-1, 2-propanediol.

**Scheme 4.** Base catalyzed transesterification of the 3-chloro-1,2-propylenecarbonate to DMC and 3-Chloro-1, 2-propanediol.

After the interaction of the equivalent amounts of DBU and 3-chloro-1,2-propylenecarbonate in methanol for 30 min, it was observed that the expected products such as DMC and 3-chloro-1, 2-propanediol formed in the reaction mixture (Figure 5b). However, besides the signal for these chemical species, new signals at 44.3, 52.4, and 62.1 ppm were also observed. The reaction mixture was stirred for various reaction times such as 2, 6 and 18 h and it was observed that these newly observed signals belong to unknown chemical species, the amounts of which gradually increased. Simultaneously, the signal belonging to 3-chloro-1, 2-propanediol steadily decreased as the reaction time increased. Besides that, it was also observed that the chemical shifts for the carbon atoms of DBU, especially at positions six and nine, respectively, were shielded, whereas the carbon atom at position seven became de-shielded under the given reaction period.

Considering the changes in the chemical shifts for the carbon atoms in the DBU molecule, it is evident that the [DBUH]⁺ cation is gradually forming in the reaction mixture.

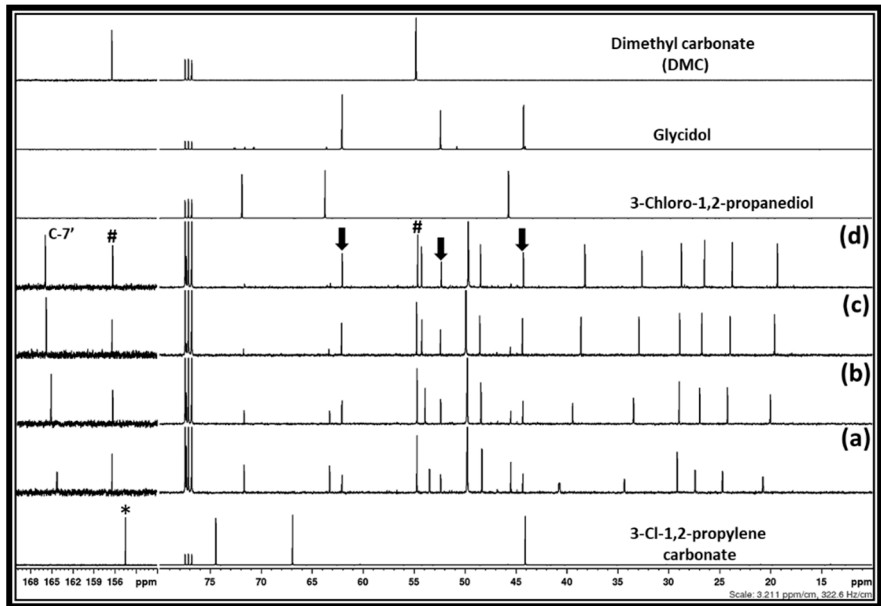

**Figure 5.** ¹³C NMR spectra for the room temperature and DBU catalyzed transesterification of 3-chloro-1,2-propylenecarbonate in methanol, (**a**) 30 min, (**b**) 2 h, and (**c**) 6 h and, (**d**) 18 h (Downward arrows and # sign showed carbon atoms belongs to the glycidol and DMC, respectively).

It was previously reported that the synthesis of epoxides such as epichlorohydrin and glycidol from the hydro-chlorinated analogs of glycerol such as 1,3-dichloro-2-propanol and 3-Chloro-1, 2-propanediol, respectively, is a base-catalyzed reaction where an equivalent interaction of the base with either of these chlorinated species results in the formation of the corresponding epoxides [5,7]. In the present work, since the transesterification of 3-chloro-1,2-propylenecarbonate was carried out with an equivalent amount of DBU, the possibility is that the in situ-formed 3-chloro-1, 2-propanediol can transform further to an oxiranic function-comprising molecule, i.e., glycidol through the release of a Cl atom with a DBU base (Scheme 5). To confirm the glycidol formation, the NMR spectra of the reaction mixture after the 18 h and commercially available glycidol were compared and it was observed that identical signals related to glycidol (shown by downward arrow) were observed. Since the signal related to the [DBUH]⁺ cation was also observed, this also confirmed that the formation of glycidol has occurred through the formation of [DBUH][Cl] salt in the reaction composition. Hence, the base-catalyzed transesterification of 3-chloro-1,2-propylenecarbonate in methanol results in the formation of DMC and glycidol along with [DBUH][Cl]. Overall, the dechlorination of the glycerol-derived 1,3-dichloro-2-propanol has occurred during the synthesis of DMC as well as glycidol, which not only facilitates the uptake of $CO_2$ but also allowed for the synthesis of industrially important value-added chemicals. Besides that, the DBU molecule not only assisted in the efficient $CO_2$ capture and served as a base catalyst in DMC synthesis but also performed as a reservoir for the chloride ion through the formation of its non-volatile and thermally stable chloride salt.

The synthesis of DMC and glycidol from 3-chloro-1,2-propylenecarbonate was further carried out at higher temperatures such as 35 and 50 °C, where the rate of formation of glycidol increased with the temperature and the complete conversion of the in situ-formed 3-Chloro-1,2-propanediol to glycidol took place in 2 h and 30 min, respectively (supporting information Figure S4a,b). Hence, in this synthesis, the applied temperatures significantly influenced glycidol synthesis levels, whereas the rate of DMC formation remained unaltered.

**Scheme 5.** Synthesis of glycidol and [DBUH][Cl] salt from 3-chloro-1,2-propanediol and DBU.

Using the distillation method, 92% of DMC was recovered along with methanol from the reaction mixture, whereas the remaining glycidol and [DBUH][Cl] salt were separated using solvent extraction. In the case of solvent extraction initially, 2-Me-THF was added in a mixture of [DBUH][Cl] salt and glycidol in order to remove glycidol selectively from the mixture. However, after the addition of 2-Me-THF, no solid [DBUH][Cl] salt precipitated out from the mixture. On the other hand, the turbid solution was obtained after 1 h and the transparent viscous liquid settled at the bottom of the flask. The separation of [DBUH][Cl] from the glycidol is not possible, probably as a result of the formation of a deep eutectic mixture through hydrogen bond acceptor (HBA) and hydrogen bond donor (HBD) interactions. In this case, considering previous reports regarding the compositions of various deep eutectic solvents (DES), the chloride anion containing ionic liquids such as choline chloride and a hydroxyl group comprised of molecules such as glycerol, ethylene glycol, etc., performed as a HBD and HBA, respectively to form a DES [30]. Sato et al. also showed that a similar hydrogen bonding interaction between the –OH group of glycidol and the Cl$^-$ anion of the quaternary alkylammonium salt was established and resulted in the formation of a binding complex [31]. Therefore, the hydrogen bonding interaction possibly does not allow the precipitation of [DBUH][Cl] salt in 2-Me-THF. In order to trigger the separation of these two chemical species, the brine solution was added to their mixture, followed by the addition of 2-Me-THF to extract glycidol. In this case, NaCl in the brine solution possibly disturbed the hydrogen bonding between [DBUH][Cl] salt and glycidol and allowed the transfer of the latter to the 2-Me-THF phase. After the evaporation of 2-Me-THF from the organic phase, an 89% recovery of the pure form of the glycidol was achieved. The 13C NMR spectra of the recovered glycidol is shown in Figure 6. Water was removed from the aqueous phase and dry methanol was added to precipitate and separate NaCl from the [DBUH][Cl] salt. The alkaline alcoholic solution (NaOH in methanol) was further used to recover the molecular DBU from its chloride salt. In this context, 83% of the pure form of the DBU was obtained and the spectra of the recovered DBU is shown in supporting information Figure S5.

Hence, in this overall reaction approach, the hydro-chlorinated derivative of glycerol, i.e., 1,3-dichloro-2-propanol, was further used for the $CO_2$ capture and its further valorization was demonstrated to synthesize DMC along with the glycidol. The specialty of this work is that DBU superbase was applied for various tasks where it performed as an efficient, selective, and recoverable base catalyst along with $CO_2$ capturing as well as a dechlorinating agent in the synthesis. Apart from this, we introduce a new alternative and simultaneous synthetic approach to produce DMC and glycidol compared to the existing various individual catalytic approaches. In terms of the solvent system, 2-Me-THF emerged as a new and renewable solvent media for the $CO_2$ activation to value-added chemicals, which was previously limited to $CO_2$ capture in the form of expanded liquids.

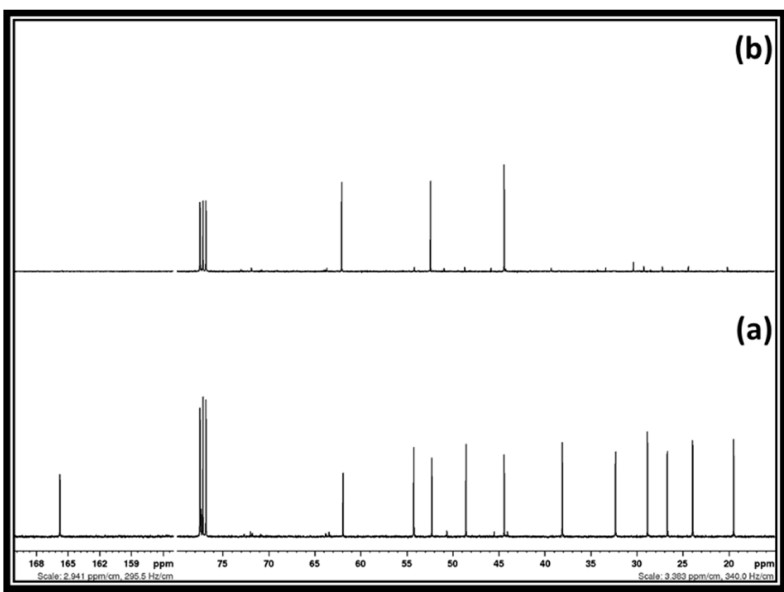

**Figure 6.** $^{13}$C NMR spectra: (**a**) [DBUH][Cl] salt and glycidol after removal of DMC and methanol and, (**b**) recovered glycidol.

## 4. Conclusions

The glycerol-derived 1,3-dichloro-2-propanol as well as $CO_2$ was successfully valorized for the synthesis of an industrially important dimethyl carbonate and glycidol integrated process approach under mild reaction conditions. The synthesis of 3-chloro-1,2-propylenecarbonate from 1,3-dichloro-2-propanol was carried out in the DBU superbase-triggered $CO_2$ capture process followed by a cyclization approach at room temperature, whereupon the formation of cyclic carbonate was confirmed with both one- and two-dimensional NMR spectroscopy techniques. Amongst the applied solvents, upon the use of DMSO, 69% of 3-chloro-1,2-propylenecarbonate was recovered, on the other hand, 2-Me-THF emerged as a more efficient solvent system compared to DMSO, whereupon a 97% recovery was achieved without the use of solvent extraction. The synthesized 3-chloro-1,2-propylenecarbonate was further explored for the base-catalyzed transesterification to synthesize DMC in methanol, whereupon in situ-formed 3-chloro-1, 2-propanediol simultaneously converted to glycidol as a result of its equivalent interaction with DBU superbase. In this integrated process for the synthesis of DMC and glycidol, DBU superbase performed versatile tasks where it participated in the $CO_2$ activation and base-catalyzed alcoholysis process along with an efficient dechlorinating agent through the formation of a thermally stable [DBUH][Cl] salt. In the case of the alcoholysis of 3-chloro-1,2-propylenecarbonate to DMC and glycidol, the rate of the formation of DMC was not influenced by the applied temperature, whereas the rate of the formation of glycidol linearly increasing with the applied temperature. In this case, 3-chloro-1,2-propanediol was completely converted to glycidol in 30 min at 50 °C while at room temperature. Some of the 3-chloro-1,2-propanediol remained unreacted even after 18 h. The 93% of DMC along with methanol was recovered from the reaction mixture by evaporation, whereas 89% of glycidol was obtained from the mixture of [DBUH][Cl] salt using the brine solution and 2-Me-THF involved solvent extraction. In this case, the brine solution facilitated the separation of glycidol from the [DBUH][Cl] salt. The DBU superbase was also obtained with 83% recovery from the [DBUH][Cl] salt following a neutralization approach. Hence, in this process, the sustainable valorization of $CO_2$ along with a glycerol derivative such as 1,3-dichloro-2-propanol was demonstrated using recoverable DBU superbase and a renewable solvent, 2-Me-THF. This new reaction pathway can be further explored for the synthesis of other dialkyl carbonates along with glycidol, where, in this case, various types of alcohols can be utilized.

**Supplementary Materials:** The following are available online at https://www.mdpi.com/article/10.3390/cleantechnol3040041/s1, Figure S1: Calibration curve for the quantification of recovered dimethyl carbonate (Gas chromatography method). Figure S2: [1]H NMR spectra of the (a) 1,3-dichloro-2-propanol, (b) DBU, and (c) Reaction mixture after equivalent interaction of 1,3-dichloro-2-propanol, DBU, and $CO_2$ in DMSO (NMR analysis with capillary filled with $D_2O$). Figure S3: [1]H-[13]C HSQC (a) and [1]H-[1]H COSY (b) NMR spectra of the reaction mixture after equivalent interaction of 1,3-dichloro-2-propanol, DBU, and $CO_2$ in DMSO. Figure S4a: [13]C NMR spectra for the DBU catalyzed transesterification of 3-chloro-1,2-propylenecarbonate in methanol at 35 °C, (a) 30 min, (b) 1 h, and (c) 2 h. (Downward arrows and # sign showed carbon atoms belongs to the glycidol and DMC, respectively). Figure S4b: [13]C NMR spectra for the DBU catalyzed transesterification of 3-chloro-1,2-propylenecarbonate in methanol at 50 °C, (a) 15 min and (b) 30 min. (Downward arrows and # sign showed carbon atoms belongs to the glycidol and DMC, respectively). Figure S5: [13]C NMR spectra (a) [DBUH Cl] salt and (b) recovered DBU.

**Author Contributions:** Conceptualization, S.K.; methodology, S.K.; validation, S.K. and G.S.; resources, S.K.; data curation, S.K. and G.S.; writing—original draft preparation, S.K.; writing—review and editing, S.K., G.S. and J.-P.M.; supervision, J.-P.M.; project administration, S.K. and J.-P.M.; funding acquisition, J.-P.M. All authors have read and agreed to the published version of the manuscript.

**Funding:** This research received no external funding.

**Institutional Review Board Statement:** Not applicable.

**Informed Consent Statement:** Not applicable.

**Acknowledgments:** This work is part of activities of the Technical Chemistry, Department of Chemistry, Chemical-Biological Centre, Umeå University, Sweden as well as the Johan Gadolin Process Chemistry Centre at Åbo Akademi University in Finland. The Swedish Research Council (Drn: 2016-04090), Bio4Energy programme, Kempe Foundations and Wallenberg Wood Science Center under auspices of Alice and Knut Wallenberg Foundation are gratefully acknowledged for funding this project.

**Conflicts of Interest:** The authors declare no conflict of interest.

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
