# Peer review of "Integrated and Metal Free Synthesis of Dimethyl Carbonate and Glycidol from Glycerol Derived 1,3-Dichloro-2-propanol via CO2 Capture"

_cleantechnol, doi:10.3390/cleantechnol3040041_

Round 1

Reviewer 1 Report

  1. Khokarale and J-P. Mikkola et al. have succeeded in the integrated synthesis of the industrially important compounds dimethyl carbonate and glycidol under mild reaction conditions using 1,3-dichloro-2-propanol derived from glycerol and CO2. The CO2 recovery process using DBU superbase and the formation of cyclic carbonates were confirmed by 1D and 2D NMR spectroscopy. The title reaction uses recoverable base and solvent and has potential industrial applications. The paper is very well written, and the compound data is sufficient. My evaluation is that the paper is publishable with minor scientific revisions.

  • Please correct some of the incorrect naming of the compounds throughout the text.

For example, "1, 2 halohydrins" should be corrected to "1,2-halohydrin".

‘2-chloro-1, 3-propanol’ to ‘2-chloro-1,3-propanol’

L140-, ’2-Cl-propylele’ to ‘2-Cl-propylene’

Author Response

Khokarale and J-P. Mikkola et al. have succeeded in the integrated synthesis of the industrially important compounds dimethyl carbonate and glycidol under mild reaction conditions using 1,3-dichloro-2-propanol derived from glycerol and CO2. The CO2 recovery process using DBU superbase and the formation of cyclic carbonates were confirmed by 1D and 2D NMR spectroscopy. The title reaction uses recoverable base and solvent and has potential industrial applications. The paper is very well written, and the compound data is sufficient. My evaluation is that the paper is publishable with minor scientific revisions.

Please correct some of the incorrect naming of the compounds throughout the text.

Point 1: For example, "1, 2 halohydrins" should be corrected to "1,2-halohydrin".

‘2-chloro-1, 3-propanol’ to ‘2-chloro-1,3-propanol’

L140-, ’2-Cl-propylele’ to ‘2-Cl-propylene’

Response 1: Required changes are carried out and highlighted in the manuscript

Reviewer 2 Report

Khokarale and coworkers presented a synthetic approach for the dimethyl carbonate and glycerol from 1,3-dichloro-2-propanol and CO2 using the DBU as base. Along the way authors have compared two different solvent system to carry out these transformations, DMSO and 2-Me-THF and found out 2-Me-THF serve as better solvent over DMSO. The reactions progress and final products were confirmed by extensive 1D and 2D NMR spectroscopy. The work has been carried out professionally and conclusions are supported with proper experimental findings.

I would like to recommend acceptance of the article for the publication in the journal "Clean technologies" after addressing below comments.

  1. Line 26: DBU doesn't serve as chlorine reservoir (as claimed) however it serve as chloride ion reservoir (line 382 too).
  2. Fig 1: In case of 2-chloropropylene carbonate and DMC one of the carbons has been depicted with * and # respectively, any specific reason? I couldn't find reference to it in text.
  3. Line 158: washed with 15 mL, with is missing in the sentence?
  4. Line 162-163: Ethyl acetate was .... vacuum drying. Doesn't sound scientifically appropriate. Replace with like " The organic phase was concentrated on rotary evaporator to obtain 2-chloropropylene carbonate after drying over anhydrous sodium sulfate or magnesium sulfate" (I assume authors dried organic phase before concentration under reduced pressure).
  5. Line 163: Typo, replace 'propylele' with 'propylene' and on the line 214, 275, 277, 279, 288, 291, 292, 293, 295, 301, 320, 323, 327, 332 etc. 
  6. Line 197: The boiling point of methanol is 64 deg C then it won't reflux at 50 deg C. Did author use different reaction temperature?
  7. Line 210-227: The chemical name 1,3-dichloro-2-propanol was written inconsistently in the same paragraphy. Be consistent throughout the manuscript.
  8. Line 219: dimenational??? (line 247)
  9. Line 239: what is SIL?
  10. Line 303: Is it solubility of CO2 in 2-Me-THF and water? Please rewrite the sentence.
  11. Line 393: Is it evaporation method or distillation method?
  12. Line 414-419: It is unclear to me the separation of NaCl and HCl salt of DBU using anhydrous methanol. Does sodium chloride get separated from DBU salt because DBU salt is soluble in anhydrous methanol. Which alkaline alcoholic solution has been used and how did it help to recover DBU? Please provide details. 

Author Response

Reviewer 2

Khokarale and coworkers presented a synthetic approach for the dimethyl carbonate and glycerol from 1,3-dichloro-2-propanol and CO2 using the DBU as base. Along the way authors have compared two different solvent system to carry out these transformations, DMSO and 2-Me-THF and found out 2-Me-THF serve as better solvent over DMSO. The reactions progress and final products were confirmed by extensive 1D and 2D NMR spectroscopy. The work has been carried out professionally and conclusions are supported with proper experimental findings.

I would like to recommend acceptance of the article for the publication in the journal "Clean technologies" after addressing below comments.

Point 1: Line 26: DBU doesn't serve as chlorine reservoir (as claimed) however it serve as chloride ion reservoir (line 382 too).

Response 1: Required changes are carried out and highlighted in the manuscript

Point 2: In case of 2-chloropropylene carbonate and DMC one of the carbons has been depicted with * and # respectively, any specific reason? I couldn't find reference to it in text.

Response 2: The purpose of use of these signs are described in the caption of figure 1.

Point 3: Line 158: washed with 15 mL, with is missing in the sentence?

Response 3: The sentence is corrected.

Point 4: Line 162-163: Ethyl acetate was .... vacuum drying. Doesn't sound scientifically appropriate. Replace with like " The organic phase was concentrated on rotary evaporator to obtain 2-chloropropylene carbonate after drying over anhydrous sodium sulfate or magnesium sulfate" (I assume authors dried organic phase before concentration under reduced pressure).

Response 4: The sentence is corrected.

Point 5: Line 163: Typo, replace 'propylele' with 'propylene' and on the line 214, 275, 277, 279, 288, 291, 292, 293, 295, 301, 320, 323, 327, 332 etc.

Response 5: The sentences are corrected.

Point 6: Line 197: The boiling point of methanol is 64 deg C then it won't reflux at 50 deg C. Did author use different reaction temperature?

Response 6: The ‘reflux’ word has removed now, since 50 oC temperature has been used for the dissolution.   

Point 7: Line 210-227: The chemical name 1,3-dichloro-2-propanol was written inconsistently in the same paragraphy. Be consistent throughout the manuscript.

Response 7: The sentences are corrected. The 1,3-dichloro-2-propanol was written in a similar way in all the manuscript.

Point 8: Line 219: dimenational??? (line 247)

Response 8: Word is corrected now.

Point 9. Line 239: what is SIL?

Response 9: The long form of the word ‘SIL’ is already mentioned in the line 103 in the manuscript therefore it is not repeated further.  

Point 10: Is it solubility of CO2 in 2-Me-THF and water? Please rewrite the sentence.

Response 10: The sentence is corrected and re-written.

Point 11: Is it evaporation method or distillation method?

Response 11: The sentence is corrected.

Point 12: Line 414-419: It is unclear to me the separation of NaCl and HCl salt of DBU using anhydrous methanol. Does sodium chloride get separated from DBU salt because DBU salt is soluble in anhydrous methanol. Which alkaline alcoholic solution has been used and how did it help to recover DBU? Please provide details.

Response 12: Yes, in this case dry methanol was used to separate the DBU salt from NaCl. In this case, the DBU salt is soluble in methanol while NaCl not.

The alkaline alcoholic solution comprised of dissolved NaOH in methanol was used for the recovery of DBU from DBU salt. The detail recovery study is included in the material and method section. Also related short information included in the manuscript.    

Reviewer 3 Report

At the present work, authors have provided a sustainable integrated two-step synthetic protocol for the synthesis of dimethyl carbonate and glycidol starting from 1, 3-dichloro-2-propanol via DBU promoted-CO2 capture. Every reaction step has been extensively studied by 1H-NMR and 13C-NMR, and the corresponding intermediates and products identified. The recovery of the 2-Cl-propylene carbonate intermediate and the products has been quantified. The DBU base used in the process is regenerated in and additional step and its recovery quantified.

Novelty: A very similar synthetic methodology than that employed in the first step (synthesis of 2-Cl-propylene carbonate) has been developed by the authors for other similar substrates in a previous work (reference 20 in the text). Moreover, the synthesis of 2-Cl-propylene carbonate from 1, 3-dichloro-2-propanol has been previously reported using K2CO3 as base and PEG-400 as solvent (Aust. J. Chem. 2009, 62, 917-920). On the other hand, the synthesis of dimethyl carbonate from 2-Cl-propylene carbonate has been reported (US patent US 20170107170 A1). Hence, the novelty of this work lies in the obtainment of glycidol as product together with the dimethyl carbonate, which presents interest, and the benefits of the synthetic protocol since the point of view of sustainability.   

In general, the text and formulas require an extensive checking in order to avoid large amount of mistakes (like “propylele” instead “propylene”) and grammatical errors. Moreover, the text is too much dense in several parts of the manuscript. Overall, the section 2 Materials and Methods requires an important check.    

Author Response

Reviewer 3

At the present work, authors have provided a sustainable integrated two-step synthetic protocol for the synthesis of dimethyl carbonate and glycidol starting from 1, 3-dichloro-2-propanol via DBU promoted-CO2 capture. Every reaction step has been extensively studied by 1H-NMR and 13C-NMR, and the corresponding intermediates and products identified. The recovery of the 2-Cl-propylene carbonate intermediate and the products has been quantified. The DBU base used in the process is regenerated in and additional step and its recovery quantified.

Novelty: A very similar synthetic methodology than that employed in the first step (synthesis of 2-Cl-propylene carbonate) has been developed by the authors for other similar substrates in a previous work (reference 20 in the text). Moreover, the synthesis of 2-Cl-propylene carbonate from 1, 3-dichloro-2-propanol has been previously reported using K2CO3 as base and PEG-400 as solvent (Aust. J. Chem. 2009, 62, 917-920). On the other hand, the synthesis of dimethyl carbonate from 2-Cl-propylene carbonate has been reported (US patent US 20170107170 A1). Hence, the novelty of this work lies in the obtainment of glycidol as product together with the dimethyl carbonate, which presents interest, and the benefits of the synthetic protocol since the point of view of sustainability.   

Point 1: In general, the text and formulas require an extensive checking in order to avoid large amount of mistakes (like “propylele” instead “propylene”) and grammatical errors. Moreover, the text is too much dense in several parts of the manuscript. Overall, the section 2 Materials and Methods requires an important check.

Response 1: Required changes are carried out and highlighted in the manuscript